# The effect of storage conditions on microbial communities in stool

**Kristien Nel Van Zyl**[1]*, **Andrew C. Whitelaw**[1,2,3], **Mae Newton-Foot**[1,2]

**1** Division of Medical Microbiology, Department of Pathology, Stellenbosch University, South Africa,
**2** National Health Laboratory Service, Tygerberg Hospital, Cape Town, South Africa, **3** African Microbiome Institute, Stellenbosch University, South Africa

* knvz@sun.ac.za

**Data Availability Statement:** Data are available from Dryad: (Dryad DOI: 10.5061/dryad. zw3r2284t).

**Funding:** This study was supported by grants funded through the NHLS Research Trust of South

## Abstract

Microbiome research has experienced a surge of interest in recent years due to the advances and reduced cost of next-generation sequencing technology. The production of high quality and comparable data is dependent on proper sample collection and storage and should be standardized as far as possible. However, this becomes challenging when samples are collected in the field, especially in resource-limited settings. We investigated the impact of different stool storage methods common to the TB-CHAMP clinical trial on the microbial communities in stool. Ten stool samples were subjected to DNA extraction after 48-hour storage at -80˚C, room temperature and in a cooler-box, as well as immediate DNA extraction. Three stool DNA extraction kits were evaluated based on DNA yield and quality. Quantitative PCR was performed to determine the relative abundance of the two major gut phyla Bacteroidetes and Firmicutes, and other representative microbial groups. The bacterial populations in the frozen group closely resembled the immediate extraction group, supporting previous findings that storage at -80˚C is equivalent to the gold standard of immediate DNA extraction. More variation was seen in the room temperature and cooler-box groups, which may be due to the growth temperature preferences of certain bacterial populations. However, for most bacterial populations, no significant differences were found between the storage groups. As seen in other microbiome studies, the variation between participant samples was greater than that related to differences in storage. We determined that the risk of introducing bias to microbial community profiling through differences in storage will likely be minimal in our setting.

## Introduction

The collection, transport and storage of stool samples are major challenges faced by gut microbiome researchers in resource-limited settings. Immediate freezing of stool samples at -80˚C preserves the microbiome composition when compared to the true gold standard of immediate DNA extraction [1–4]. However, cold-chain storage and the addition of DNA-stabilizing agents are not always feasible when samples are collected in the field. Furthermore, thawing of

Africa (MNF received GRANT004_ 94632 and GRANT004_94679) and the Harry Crossley Foundation (KN received a HCF grant). The funders had no role in study design, data collection and analysis, decision to publish, or preparation of the manuscript.

**Competing interests:** The authors have declared that no competing interests exist.

samples during transport may significantly reduce DNA integrity thereby influencing the accuracy of downstream microbiome analysis [2,5].

In our study setting, stool samples collected as part of the TB-CHAMP clinical trial (https://doi.org/10.1186/ISRCTN92634082) will be used for both microbiome analysis and culture-based analysis. Given the limited resources, it is not feasible to split the stool samples for the two approaches. This makes the addition of most DNA stabilizing agents impossible, due to their bactericidal effect. The samples should ideally be collected and delivered to the laboratory within 24 hours; however, some samples may also be collected from participants' homes, where there may not be access to a freezer. Therefore, the samples may be subjected to longer periods in a cooler box on ice or even at room temperature. Several studies have aimed to assess the effect of temperature and time on microbial communities in stool; however, the methods were variable and the results (unsurprisingly) were conflicting [1,3,5–9]. The uniqueness of microbiota composition and influencing factors in different settings, along with lack of standardization of study procedures emphasize the need to determine the possible biases introduced by storage and transport methods in all microbiome studies.

For these reasons, we investigated how different storage methods common to this study may affect the microbial communities in stool. Immediate DNA extraction (the gold standard technique) was compared to 48-hour storage with no additives at -80°C, room temperature (between 20°C and 30°C) and on an ice brick in a cooler box to simulate real-life field collection scenarios common to the trial.

While next-generation sequencing-based assays remain the gold standard for microbiome analysis, they are relatively expensive and time-consuming. We performed quantitative PCR (qPCR) as an alternative to sequencing [10–12], to provide a rapid broad overview of the microbiome, which is ideal for a pilot study such as this one. By targeting specific bacterial groups with qPCR and comparing them to the total number of bacteria present in a sample (as determined by Universal Eubacteria primers) their abundance could be determined, and changes could be observed quickly. The phyla Bacteroidetes and Firmicutes, which comprise about 90% of the healthy human gut microbiota, and several other representative groups, namely Enterobacteriaceae, *Lactobacillus spp*. and *Bifidobacterium spp*., were targeted [12]. We also investigated the change in abundance of fungi compared to total bacteria as a marker of the effect of storage conditions on the fungal microbiota.

## Materials and methods

### Sample population and collection

Ten stool samples collected from children (<5 years) as part of the TB-CHAMP clinical trial were randomly selected for this study. The samples were collected in a standard specimen container without preservative and delivered to the laboratory for processing on the day of collection. Stool samples were homogenized to prevent intra-sample variability caused by sub-sampling different microenvironments of the stool [12]. Homogenization was performed by mixing with the sterile spoon attached to the inside of the collection tube lid. Aliquots from each sample were subjected to either immediate DNA extraction, 48-hour storage with no additives at -80°C, 48-hour storage at room temperature (between 20°C and 30°C), or 48-hour storage on an ice brick in a cooler box. Temperature fluctuations were monitored by recording the ambient temperatures at room temperature and in the cooler box over a 48-hour period. Written consent was obtained from the parents/guardians of the participants as part of the clinical trial and ethical approval granted by the Stellenbosch University Human Research Ethics Committee (SU-HREC) (M16/02/009 and S18/02/031) and the Medicines Control Council of South Africa (20160128).

## DNA extraction and quality analysis

Prior to DNA extraction, the aliquots were homogenised a second time, to avoid variability caused by changes in the microenvironments after 48 hours of storage. For the first five samples, three different commercially available stool DNA extraction kits were compared: PSP® Spin Stool DNA Kit (Stratec Biomedical, Germany), ZymoBIOMICS DNA Miniprep Kit (Zymo Research, USA) and QIAamp PowerFecal DNA Isolation Kit (Qiagen, formerly moBio, Germany). The workflow is shown in Fig 1. Briefly, for each kit, DNA was extracted from 200 mg aliquots of stool from each of the four storage conditions, according to the manufacturer's instructions. Aliquots frozen at -80˚C were not thawed prior to extraction. DNA purity (A260/A280 and A260/A230), and yield (ng/µL) were determined using the BioDrop µLite spectrophotometer (BioDrop, UK) and by Qubit Fluorometric Quantitation (Invitrogen, USA) at the Central Analytical Facility (CAF) of Stellenbosch University. We used the Kruskal-Wallis test [13] for non-normally distributed data to determine whether the different extraction kits or storage methods influenced the DNA yield significantly ($p < 0.05$). DNA degradation was assessed by gel electrophoresis. The remaining five samples were subjected to the same storage condition comparisons as described above and DNA was extracted with the kit that showed the best performance.

## Relative quantitation of microbial subpopulations

We adapted a previously described qPCR protocol [12] to determine the abundance of various microbial subpopulations, namely Bacteroidetes, Firmicutes, Enterobacteriaceae, *Bifidobacterium* spp. and *Lactobacillus* spp., relative to Eubacterial 16S rRNA gene amplification. The *Lactobacillus* primers were replaced with a pair that amplified a smaller product [14] more suitable for use with SYBR Green dye. We also amplified fungal species using ITS1 primers [15]. Amplification was performed on the Rotor-Gene Q thermocycler (Qiagen) as singleplex reactions using 1X KAPA 2G SYBR Fast Uni Kit (KAPA Biosystems), 0.2 µM of each primer (Table 1) and nuclease free water (Qiagen) in 20 µL reactions. DNA input was standardized to 30 ng per reaction and all reactions were performed in triplicate. The adapted cycling conditions for all bacterial populations were as follows: denaturation at 95˚C for 3 minutes, followed by 40 cycles of denaturation (95˚C for 5 seconds) and annealing/extension (60˚C for

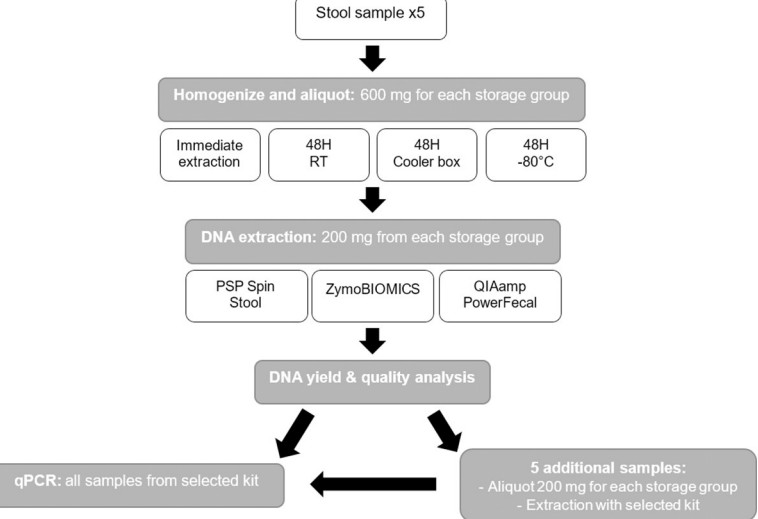

**Fig 1. Workflow for aliquoting and extraction of stool samples.** RT = Room temperature.

**Table 1. Primers used for qPCR.** All primers were synthesized by Integrated DNA technologies (USA).

| Group targeted | Primer Sequence 5' to 3' | Amplicon size | Reference |
|---|---|---|---|
| Bacteria (16S rRNA) | F: ACTCCTACGGGAGGCAGCAGT | 174–199 | [16] Walter et al., 2000 |
| | R: GTATTACCGCGGCTGCTGGCAC | | |
| Bacteroidetes | F: CGATGGATAGGGGTTCTGAGAGGA | 238 | [17] Guo et al., 2008 |
| | R: GCTGGCACGGAGTTAGCCGA | | |
| Firmicutes | F: GGAGYATGTGGTTTAATTCGAAGCA | 126 | [17] Guo et al., 2008 |
| | R: AGCTGACGACAACCATGCAC | | |
| Enterobacteriaceae | F: CATTGACGTTACCCGCAGAAGAAGC | 195 | [18] Bartosch et al., 2004 |
| | R: CTCTACGAGACTCAAGCTTGC | | |
| *Bifidobacterium* spp. | F: CGCGTCYGGTGTGAAAG | 244 | [19] Delroisse et al., 2008 |
| | R: CCCCACATCCAGCATCCA | | |
| *Lactobacillus* spp. | F: TGGAAACAGRTGCTAATACCG | 231–233 | [14] Byun et al., 2004 |
| | R: GTCCATTGTGGAAGATTCCC | | |
| Fungi (ITS1) | F: CTTGGTCATTTAGAGGAAGTAA | 260 | [15] Bellemain et al., 2010 |
| | R: GCTGCGTTCTTCATCGATGC | | |

30 seconds). The cycling conditions for fungal amplification were denaturation at 95˚C for 3 minutes, followed by denaturation (95˚C for 5 seconds) and annealing/extension at 64, 62 and 60˚C for 30 seconds for 10, 10 and 20 cycles respectively. Fluorescence was acquired to the green channel during the annealing step. The Rotor-Gene software was used to calculate the efficiency and detection threshold for each primer set using individual standard curves. The efficiencies ranged between 0.91 and 1.04 with $R^2$ values > 0.99.

## Microbial community analysis

Differences in abundance related to storage conditions were analysed using RStudio (Version 1.1.463, R version 3.6.1) and the packages tidyverse, car and lme4. Spaghetti and distribution plots were generated to show trends in abundance in the different microbial populations and samples. The data were normalised for the distribution plots by subtracting the average Universal Ct value from the target population Ct for each sample. These data were analysed using Kruskal-Wallis multiple group comparisons to investigate the differences in abundance related to storage and subject for each of the six subpopulations targeted. Bonferroni correction was performed on the basis that multiple hypotheses were tested on data from the same subjects. The raw data were also fitted to a linear mixed effect model to show the overall relationship between abundance (shown by Cycle threshold values (Ct)), storage and population, taking into account random effects introduced by the subjects.

$$Ct \sim Storage + Population + (1|Subject)$$

Correlations were deemed statistically significant at p < 0.05. We also investigated whether storage condition can influence the Firmicutes/Bacteroidetes ratio (F/B).

## Results

### DNA Extraction and Quality analysis

Based on samples from the first five subjects the PSP kit had the highest average yield, but the lowest DNA purity across all samples and storage methods (Table 2). The QIAamp kit also performed well with yields of > 20 ng/μL in all but one sample. The ZymoBIOMICS kit had substantially reduced yield and quality with samples containing more fibrous materials. The

**Table 2. Extraction kit dependent DNA quality and quantity results.** The results represent stool samples from the first five subjects extracted after storage in four different conditions. The best value in each category is indicated in bold.

| | QIAamp PowerFecal | ZymoBIOMICS | PSP Spin Stool |
|---|---|---|---|
| | **(n = 20)** | **(n = 20)** | **(n = 20)** |
| **Mean yield (ng/μL)** (min–max) | 73.04 (5.84–158.0) | 97.82 (2.62–215.2) | **135.63** (9.48–400.0) |
| **Yield < 30 ng/μL** (n) | **2 (10%)** | 5 (25%) | 4 (20%) |
| **Mean A260/A280 value** (min–max) | **1.85** (1.8–2.21) | 1.88 (1.6–2.6) | 1.99 (1.89–2.1) |
| **Mean A260/A230 value** (min–max) | **1.68** (1.04–2.11) | 1.68 (0.3–2.2) | 1.57 (0.97–2.1) |
| **Out of range A260/A280*** (n) | **1 (5%)** | 3 (15%) | 2 (10%) |
| **Out of range A260/A230**** (n) | **4 (20%)** | 5 (25%) | 8 (40%) |

* Acceptable A260/A280 range defined as 1.8–2.0.

** Acceptable A260/A230 range defined as 1.4–2.2. This is wider that the traditional range of 2.0–2.2; in our setting Illumina sequencing has been successful within this range.

QIAamp extractions had the highest overall quality, with a mean yield that was slightly lower than the other kits, but still sufficient for downstream molecular experiments such as sequencing. No significant differences in DNA yield were detected for either the kit (p = 0.5142) or storage method (p = 0.1814) using the Kruskal-Wallis test. The average DNA purity (A260/280) of the QIAamp kit was 1.85, with all but one sample within the ideal range (1.8–2.0).

No significant signs of DNA degradation were detected by gel electrophoresis for any kit and all samples had large genomic weight DNA, with minimal smearing. Of the extraction kits, the PSP kit had the most samples with some evidence of smearing. For these reasons, extractions from the QIAamp PowerFecal DNA Isolation Kit were selected for further analyses. The quality ranges above were defined to show the suitability of each kit to extract DNA for use in next-generation sequencing; however, due to the robustness of PCR, we included all extractions from this kit for further analysis in this study.

## Comparison of storage conditions

There was little fluctuation in room temperature over the 48 hours, in contrast to in the cooler box (Fig 2), where a rapid decline in temperature to just above 0˚C was observed within the

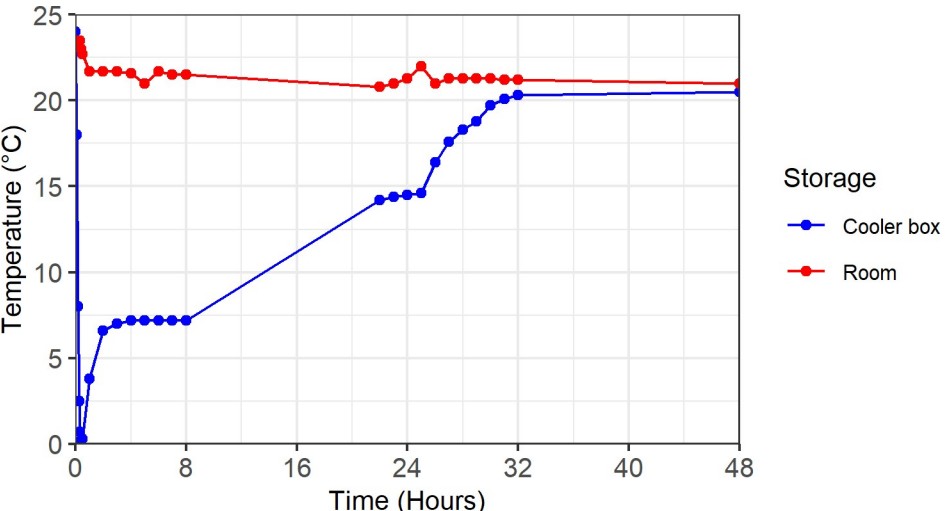

**Fig 2. Temperature fluctuations at room temperature and in the cooler box.**

first 2 hours, followed by a gradual increase of 1 degree per hour until reaching 7˚C where it remained stable until 8 hours. Over the following 24 hours, the temperature in the cooler box slowly increased until stabilizing just under room temperature until 48 hours.

## Microbial community analysis

Each of the six microbial subpopulations we targeted were detected in all but one extracted sample. The extracted DNA from the cooler box group for subject 3 (3C) was not within the detection limit for the fungal qPCR and was therefore excluded from the fungal qPCR analysis. The bacterial populations in the frozen (-80˚C) group most closely resembled the immediate extraction group; more variation was seen in the room temperature and cooler-box groups. Less abundant populations were more sensitive to differences in storage and showed more variation overall (S1 Fig). Still, with few exceptions, samples from the same subject tended to cluster together within a population group, regardless of storage method. This is demonstrated in the Enterobacteriaceae and Bifidobacteria groups, where clusters of colour (representing the subject) are seen (Fig 3).

No significant differences were detected between storage groups for any of the microbial subpopulations targeted, but subject related differences were statistically significant for each population group (Fig 3). The linear-mixed effect model showed no significant differences between the storage conditions (p = 0.1655), confirming that variation was accounted for by population- (p = < 0.00001) and random effects introduced by subject specific characteristics. The population with the least amount of variation was the Firmicutes, followed by Bacteroidetes. The Enterobacteriaceae and Fungi showed the most variation across all samples, followed by the Bifidobacteria. In these populations, it is clear that subject specific characteristics are associated with more variation in abundance than the storage conditions are. Populations that were less abundant overall, such as Fungi and Enterobacteriaceae, appeared to be more sensitive to changes in storage. In concordance with our other results, the Firmicutes/Bacteroidetes (F/B) ratio also varied more between subjects than between storage methods (Fig 4). In samples where the F/B ratio was altered, it was more often than not driven by an increase in Firmicutes, rather than a decline in Bacteroidetes, especially in the room temperature and cooler box storage groups.

## Discussion

We compared three commercially available stool DNA extraction kits and found that they generally produced sufficient yields for sequencing purposes. This is likely due to the fact that they all include a bead-beating step, which has been shown to significantly improve DNA yield from stool [20]. The QIAamp kit performed the best, although not by a large margin. This is not surprising, as similar MoBio kits (now marketed through Qiagen) have been used for DNA extraction by leaders in the field, including the Human Microbiome Project [21]. The QIAamp PowerFecal DNA Isolation Kit tested in this study and other related kits, such as the QIAamp Fast Stool kit (Qiagen), have also been found to be effective at extracting fungal DNA when they are combined with a bead-beating step [22], which demonstrates that parallel mycobiome analysis will be possible from the same DNA extracts. Choosing the correct kit is dependent on the samples, aim and resources available for the study. It is vital to use only one DNA extraction method for all samples, and to be careful when comparing microbiome results when different extractions were used [23–25]. It is also advisable to work in as sterile a fashion as possible and to randomly extract samples to control for contamination and environmental biases introduced through the extraction kit [26].

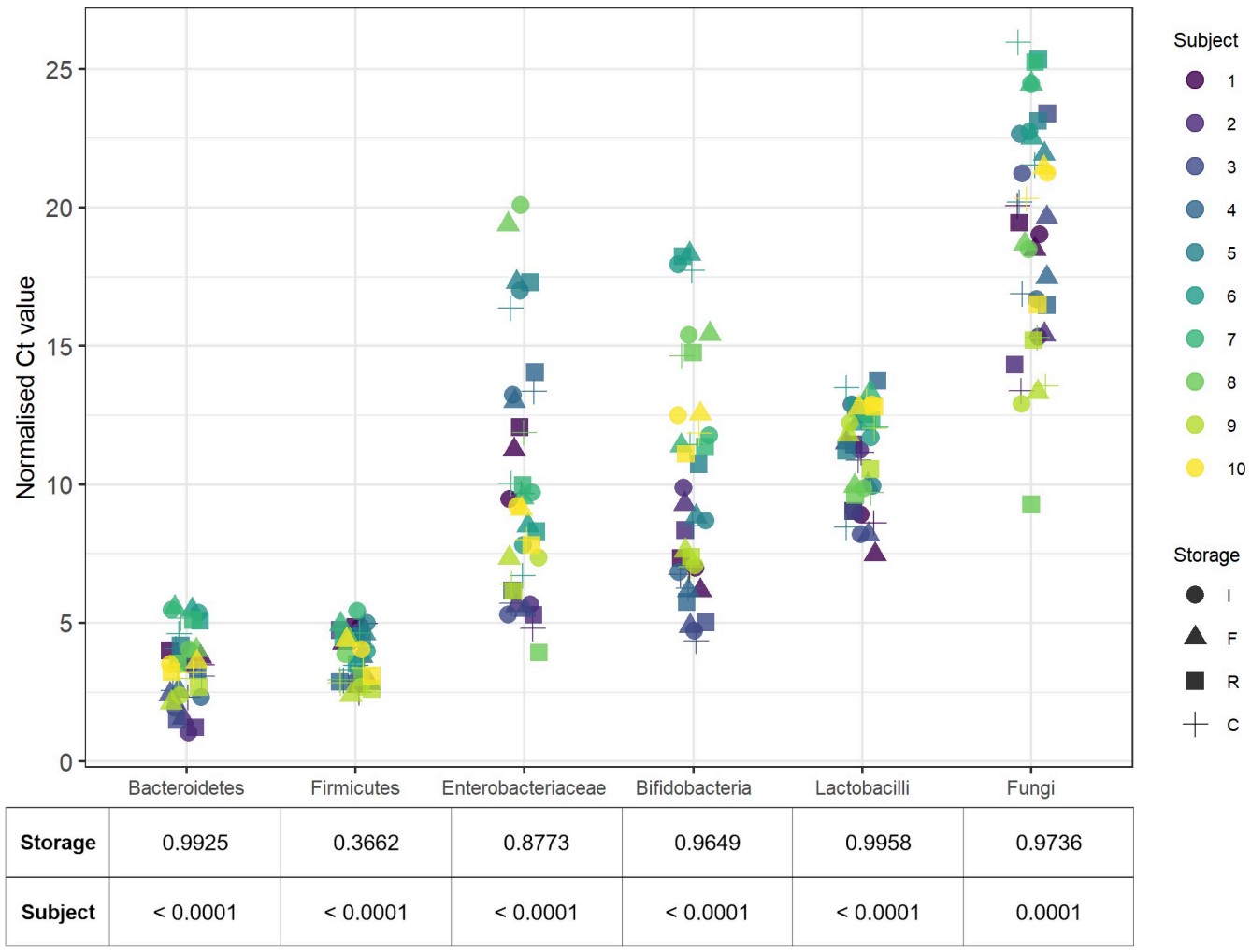

| | Bacteroidetes | Firmicutes | Enterobacteriaceae | Bifidobacteria | Lactobacilli | Fungi |
|---|---|---|---|---|---|---|
| **Storage** | 0.9925 | 0.3662 | 0.8773 | 0.9649 | 0.9958 | 0.9736 |
| **Subject** | < 0.0001 | < 0.0001 | < 0.0001 | < 0.0001 | < 0.0001 | 0.0001 |

**Fig 3. Subject specific characteristics are associated with more variation in the abundance of microbial populations than storage conditions.** A lower Ct value represents a higher abundance. Kruskal-Wallis group comparison p-values are shown below the figure. The corrected p-value for significance = 0.004, according to the Bonferroni method. Ct = Cycle threshold; I = Immediate; F = Frozen; R = Room temperature; C = Cooler box. Samples 2C and 3C had a DNA input concentration < 30ng.

The results from the microbial analysis support previous findings that storage at -80°C is equivalent to the gold standard of immediate DNA extraction [1–4]. Stool samples that are not kept in preservatives should therefore be frozen as soon as possible, and not thawed until DNA extraction. We determined that the differences in relative abundance were greater between subjects than between storage methods; this is comparable to the findings of other published studies [2,3]. While differences in abundance for all taxa was shown to be significantly related to subjects, this study was not designed to determine the drivers of differences, such as age, antibiotic exposure and diet; however, this will be investigated in a planned larger sequencing-based study from the same population. We found no significant differences in the abundance of microbes when comparing the different storage methods to immediate extraction. Our findings are supported by other published studies that investigated room temperature storage for up to 72 hours [1,3,5,8,9,23]. However, some studies have also reported significant changes after as soon as 12 hours [6,7]. These contradictory findings may be due to differences in the initial relative abundance of certain populations in a sample. Choo and colleagues [6] stated

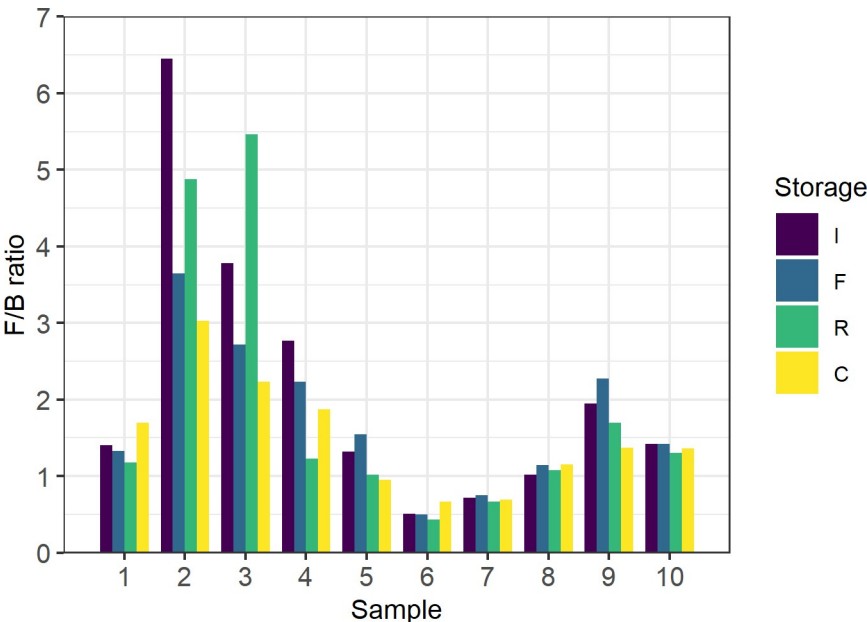

**Fig 4. The Firmicutes/Bacteroidetes ratio shows more variation between samples than between storage conditions.** This value was calculated using the normalised Ct values, which are inversely related to abundance. Therefore, the higher the F/B ratio, the more Bacteroidetes relative to Firmicutes. I = Immediate; F = Frozen; R = Room temperature; C = Cooler box.

that studies that found no significant changes due to room temperature storage reported a greater relative abundance of Bacteroidetes and Firmicutes and less than 5% Actinobacteria. The effect of room temperature storage may therefore be more severe in some study populations with higher levels of Actinobacteria; though there is not enough literature to validate this. In most cases, it is assumed that the majority of the gut microbes are mesophiles that grow best at temperatures between 20˚C and 45˚C. Therefore, the overall abundance may be expected to increase during room temperature storage with little effect on the relative abundance of each phylum. Regarding the cooler box storage conditions, we are not aware of any studies that are similar to the one presented here. We found that there were substantial changes in temperature in the cooler box storage set-up, which may have influenced the growth of certain microbial populations and thereby contributed to the variations in abundance in this group. The temperature was between 7 and 20˚C for about 24 hours, which is ideal for psychrothropic microbes which may include, among others, members of Bacilli, Enterobacteriaceae and the genera *Pseudomonas* and *Acinetobacter*. Although not substantial, we found that changes in the ratio of the two most abundant phyla was mostly driven by an increase in Firmicutes, which is in contrast to the findings of other studies that reported a decrease in Firmicutes during room temperature storage [6,7].

## Conclusion

We determined that the risk of introducing bias to microbial community profiling through differences in storage will likely be minimal in our setting. The study was limited by a small sample size and the fact that the ambient (room) temperature in the field may not be as stable as in a controlled experimental environment. Therefore, we recommend that transport time and storage conditions be recorded in similar studies in order to provide the opportunity to assess the effect of storage on the microbiome on a larger scale. While it is impractical to assess

every factor that may influence the microbiota, the collection of robust and complete metadata can help researchers identify important factors that influence the specific population of interest. Further, this study has shown that qPCR can be used to rapidly and reproducibly assess the effect of certain factors on the major bacterial populations of the gut in low-resource settings, where microbiome research is still relatively expensive. As qPCR only provides information on a limited number of bacterial populations, it remains essential to follow these studies up with next-generation sequencing to establish a more complete profile of the microbiota.

## Supporting information

**S1 Fig. The abundance of different microbial populations differ between subjects and have varying responses to different storage conditions.** This plot shows the trend in abundance for each of the subjects (the ten boxes) at the different storage conditions. A lower Ct value represents a higher abundance.
Ct = Cycle threshold; I = Immediate; F = Frozen; R = Room temperature; C = Cooler box.
Samples 2C and 3C had a DNA input concentration < 30ng.
(TIF)

## Acknowledgments

We would like to acknowledge Profs Tromp and Tabb for their guidance with the representation and statistical analysis of the data (Stellenbosch University: Division of Molecular Biology and Human Genetics; the African Microbiome Institute (AMI); and the South African Tuberculosis Bioinformatics Initiative (SATBBI), funded by the South African Medical Research Council (SAMRC) Strategic Health Innovation Partnership (SHIP) program).

## Author Contributions

**Conceptualization:** Kristien Nel Van Zyl, Andrew C. Whitelaw, Mae Newton-Foot.

**Formal analysis:** Kristien Nel Van Zyl, Mae Newton-Foot.

**Funding acquisition:** Mae Newton-Foot.

**Investigation:** Kristien Nel Van Zyl.

**Supervision:** Andrew C. Whitelaw, Mae Newton-Foot.

**Writing – original draft:** Kristien Nel Van Zyl.

**Writing – review & editing:** Kristien Nel Van Zyl, Andrew C. Whitelaw, Mae Newton-Foot.

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
