## [Decision Letter · Decision Letter 0]

11 Nov 2019

PONE-D-19-27079

The effect of storage conditions on microbial communities in stool

PLOS ONE

Dear Nel Van Zyl,

Thank you for submitting your manuscript to PLOS ONE. After careful consideration, we feel that it has merit but does not fully meet PLOS ONE’s publication criteria as it currently stands. Therefore, we invite you to submit a revised version of the manuscript that addresses the points raised during the review process.

Please address each concern raised, and explain how this concern has been addressed in the manuscript.  We need to see more statistics to support the conclusions. 

We would appreciate receiving your revised manuscript by Dec 26 2019 11:59PM. To enhance the reproducibility of your results, we recommend that if applicable you deposit your laboratory protocols in protocols.io, where a protocol can be assigned its own identifier (DOI) such that it can be cited independently in the future. For instructions see: http://journals.plos.org/plosone/s/submission-guidelines#loc-laboratory-protocols

We look forward to receiving your revised manuscript.

Kind regards,

David N Fredricks, MD

Academic Editor

PLOS ONE

Additional Editor Comments:

In addition, please explain how samples were homogenized, in detail.

2. Thank you for including the following information in your ethics statement:

"Written consent was obtained as part of the clinical trial".

We note that you included minors (age<18) in your study. Please provide additional details regarding minors' consent. In the ethics statement in the Methods and online submission information, please ensure that you have specified whether you obtained consent from parents or guardians.

Reviewers' comments:

Reviewer's Responses to Questions

**Comments to the Author**

1. Is the manuscript technically sound, and do the data support the conclusions?

Reviewer #1: Partly

2. Has the statistical analysis been performed appropriately and rigorously? 

Reviewer #1: No

3. Have the authors made all data underlying the findings in their manuscript fully available?

Reviewer #1: Yes

4. Is the manuscript presented in an intelligible fashion and written in standard English?

Reviewer #1: Yes

5. Review Comments to the Author

Reviewer #1: The authors presented findings of the stability of targeted bacteria, using qPCR, in stool samples processed and stored using different methods. The authors provide adequate rationale for conducting this study, and its findings could be important toward informing those studies that similarly desire not to use preservative agents for microbial preservation in field studies. However, there are some issues with the authors’ terminology and the statistical analyses presented that need to be addressed prior to publication in PLOS One.

Line 55-57: This sentence should be revised: 1) microbial stability should not be limited to ‘specific’ populations, and should theoretically be consistent across populations, given the same conditions/storage/extraction/processing; and 2) the data used by the authors are also ‘limited’ in both sample size (n=10) and population (children); therefore, the rationale in this sentence seems not justified.

Line 76: The authors should state if subjects were on antibiotics at the time of collection

Line 80-81: It is suggested that authors use a table or diagram to demonstrate the aliquoting of the samples and how many samples were compared within each collection method. They should also include which samples dropped out due to DNA yield/quality

Lines 125-132: Statistical analyses were insufficient to appropriately compare the stability of the microbial data between the storage methods. It is suggested that the authors use more quantitative comparisons, rather than basing their findings on the subjective figure interpretations. For example, the authors could compare the adjust mean Ct for each collected method, using linear mixed effects models, and calculate a p-value for each of the collection methods compared to the referent immediately-extracted group.

Line 133: The random effect should be stated as the subject and not the sample (since there are multiple measurements on one subject)

Line 126, 128, 143: Authors should avoid using terms as ‘cause’ and ‘effect’

Line 165-166: It is unclear what the authors are referring to

Figure 2: It is unclear what the 10 boxes represent. It also appears that many of the more abundant taxa were variable (not just the less abundant taxa)

Figure 3: This seems redundant, and it is not clear that the subject cluster together. There needs to be a statistical test to quantify variation explained by subject versus storage

Line 179: It is unclear why a single p-value is being used to compared multiple taxa abundances across the storage methods, this does not seem like the appropriate statistical test

6. PLOS authors have the option to publish the peer review history of their article (what does this mean?). If published, this will include your full peer review and any attached files.

Reviewer #1: No

---

## [Author Response · Author response to Decision Letter 0]

17 Dec 2019

To the editor and reviewers,

Our responses have been detailed fully in the "Response to Reviewers" letter. We request that the responses be read in that document, as the layout and formatting are more pliable. We have summarized those responses below:

Response to editor’s comments:

 With regards to sample homogenization and informed consent: We have expanded on both comments in the edited submission.

Response to reviewer’s comments:

 Line 55-57: 

Microbiota and what influences them are diverse and unique in different settings; therefore, it is important to study the possible influences of study procedures in each setting. We have revised the sentence to reflect this.

 Line 76: The authors should state if subjects were on antibiotics at the time of collection

• As this paper focuses on the differences related to storage relative to immediate extraction, we do not believe that it impacts the conclusions we have drawn.

• Participants were excluded if they had received ≥14 days of isoniazid or a fluroquinolone at enrolment, or if they had been treated for TB in the 12 months before. However, information regarding other antibiotics is not currently available to the authors as the trial is currently ongoing.

• Antibiotic usage data have been captured as part of the clinical trial, and have been requested from the trial data management team, but will only be made available for a larger sequencing-based study going forward.

• We have noted in the discussion section that differences in subjects, including the influence of antibiotics, will be investigated as part of a larger study going forward.

 Line 80-81: It is suggested that authors use a table or diagram to demonstrate the aliquoting of the samples and how many samples were compared within each collection method. They should also include which samples dropped out due to DNA yield/quality

We have added a workflow (Fig 1) to demonstrate the aliquoting of samples. No samples were excluded from the qPCR analysis due to quality, due to the robustness of PCR. We have included a statement in the results section to this effect.

 Lines 125-132: Statistical analyses were insufficient to appropriately compare the stability of the microbial data between the storage methods. 

Additional statistical measures have been described below and in the edited manuscript.

 Line 133: The random effect should be stated as the subject and not the sample (since there are multiple measurements on one subject)

This has been revised in the manuscript.

 Line 126, 128, 143: Authors should avoid using terms as ‘cause’ and ‘effect’

These terms have been revised in the manuscript where appropriate.

 Line 165-166: It is unclear what the authors are referring to

We have revised this sentence in the edited manuscript to reflect our meaning more clearly.

 Figure 2: It is unclear what the 10 boxes represent. It also appears that many of the more abundant taxa were variable (not just the less abundant taxa)

We have determined that Fig 3 represents our core findings most clearly and have kept it in the main text, but request to keep Fig 2 as a supplemental figure to show the variations for each subject (the ten boxes) across populations. The figure heading has been expanded to refer to the ten boxes as subjects.

 Figure 3: This seems redundant, and it is not clear that the subject cluster together. There needs to be a statistical test to quantify variation explained by subject versus storage

We have added p-values to this figure to demonstrate that statistical tests support our findings. We have described the statistical tests that generated these values below, and in the manuscript in the methods section.

 Line 179: It is unclear why a single p-value is being used to compared multiple taxa abundances across the storage methods, this does not seem like the appropriate statistical test

In consultation with the statisticians, it was felt that the population differences are inherently part of the influence and it was therefore included as a fixed effect in the linear mixed effect model. We used the entire data set, including all populations including the Universal amplification Cts (therefore, non-normalized Ct values) and indicated the p-value for differences related to storage as determined by the lme model. This model also output a p-value for population, and we have added that in the edited manuscript.

We agree with the reviewer that different taxa/populations may warrant separate statistical tests. For this reason, we decided to perform statistical testing using the normalized mean/median Ct values for each taxon. Instead of fitting linear mixed effect models for each storage group compared to the immediate as suggested by the reviewer, we performed Kruskal-Wallis multiple group comparisons. The reason for this is that linear mixed-effect models are more useful when more data is available – comparing the mean/median Ct values therefore diminishes the number of data points available.

Kind regards,

Kristien Nel Van Zyl

Division Medical Microbiology, Department of Pathology, Stellenbosch University

---

## [Editor Report · Decision Letter 1]

20 Dec 2019

The effect of storage conditions on microbial communities in stool

PONE-D-19-27079R1

Dear Dr. Nel Van Zyl,

We are pleased to inform you that your manuscript has been judged scientifically suitable for publication and will be formally accepted for publication once it complies with all outstanding technical requirements.

With kind regards,

David N Fredricks, MD

Academic Editor

PLOS ONE
---

## [Editor Report · Acceptance letter]

3 Jan 2020

PONE-D-19-27079R1 

The effect of storage conditions on microbial communities in stool 

Dear Dr. Nel Van Zyl:

I am pleased to inform you that your manuscript has been deemed suitable for publication in PLOS ONE. Congratulations! Your manuscript is now with our production department. 

With kind regards,

on behalf of

Dr. David N Fredricks 

%CORR_ED_EDITOR_ROLE%

PLOS ONE